# SelfEval: Leveraging discriminative nature of generative models for evaluation

**Sai Saketh Rambhatla**                                    *rssaketh@meta.com*
*GenAI, Meta*

**Ishan Misra**                                             *imisra@meta.com*
*GenAI, Meta*

**Reviewed on OpenReview:** *https://openreview.net/forum?id=0mGho8wrv5*

## Abstract

We present an automated way to evaluate the text alignment of text-to-image generative diffusion models using standard image-text recognition datasets. Our method, called SELFEVAL, uses the generative model to compute the likelihood of real images given text prompts, and the likelihood can be used to perform recognition tasks with the generative model. We evaluate generative models on standard datasets created for multimodal text-image discriminative learning and assess fine-grained aspects of their performance: attribute binding, color recognition, counting, shape recognition, spatial understanding. Existing automated metrics rely on an external pretrained model like CLIP (VLMs) or LLMs, and are sensitive to the exact pretrained model and its limitations. SELFEVAL sidesteps these issues, and to the best of our knowledge, is the first automated metric to show a high degree of agreement for measuring text-faithfulness with the gold-standard human evaluations across multiple generative models, benchmarks and evaluation metrics. SELFEVAL also reveals that generative models showcase competitive recognition performance on challenging tasks such as Winoground image-score compared to discriminative models. We hope SELFEVAL enables easy and reliable automated evaluation for diffusion models.

## 1 Introduction

In the past few years, generative image models have rapidly advanced and state-of-the-art text-to-image models now generate high quality realistic images. While a lot of research effort is focussed on improving these models, their evaluation has received considerably less attention. Evaluations for text-to-image models typically focus on two aspects: (1) quality of the generated image; and (2) the alignment between the generated image and the input text, *i.e.*, the 'faithfulness' of the generation. The gold standard for evaluating text-to-image models is to compare generations from pairs of models using human judgement. However, using pairwise human evaluations does not scale to lots of models or generations, making it difficult to convert them to ordinal metrics to rank models. Thus, automatic evaluations are commonly used as a proxy for comparing models.

In this work, we focus on automatic evaluations that measure the text adhering capabilities of a generative diffusion model and ask the question: can the diffusion model itself be used to measure the relatedness of an image-text pair and thus evaluate its own generations? Most works on text-to-image diffusion models focus on sampling good images given a text prompt. However, as shown in Figure 1, diffusion models can be used to estimate the conditional likelihood of an image $\mathbf{x}$ given a text prompt $\mathbf{c}$, *i.e.*, $p(\mathbf{x}|\mathbf{c})$. We propose SELFEVAL which is a practical way to estimate such likelihoods accounting

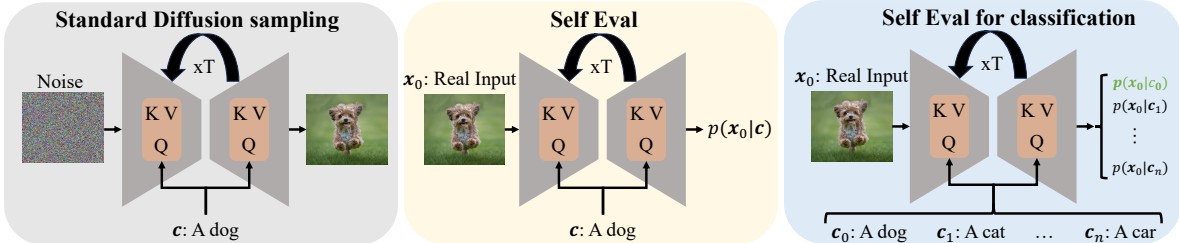

**Figure 1: Illustration of proposed method:** (Left) Starting from a noised input, the standard diffusion sampling method denoises the input iteratively to generate images from the input distribution. (Middle): SELFEVAL takes a pair (image $\mathbf{x}_0$ and conditioning $\mathbf{c}$) to estimate the likelihood $p(\mathbf{x}_0|\mathbf{c})$ of the pair in an iterative fashion. (Right): Given an image, $\mathbf{x}_0$ and $n$ captions, $\{\mathbf{c}_0, \mathbf{c}_1, \ldots, \mathbf{c}_n\}$, SELFEVAL is a principled way to convert generative models into discriminative models. We show that the classification performance of these classifiers can be used to evaluate the generative capabilities.

for numerical issues arising in standard diffusion models. We show that these likelihoods can be used directly to solve recognition tasks and evaluate the model's text-faithfulness ability.

SELFEVAL repurposes standard multimodal image-text datasets such as Visual Genome, COCO and CLEVR to measure the model's text understanding capabilities. SELFEVAL uses **ground truth** (real) image-text pairs for evaluation and computes classification accuracy making it more robust and interpretable. Our evaluation allows us to assess fine-grained aspects such as the model's ability to recognize colors, count objects *etc*. We apply our method to a wide variety of diffusion models: different types of image representations (pixel based, latent space based), different text encoders and model sizes. SELFEVAL's automatic evaluation results are in agreement with the 'gold-standard' human judgements making SELFEVAL suitable for evaluation.

Existing automated evaluations for text faithfulness of generative models rely on an external discriminative model, *e.g.*, CLIP or LLMs, to measure the 'relatedness' of the generated image to the input text. As we show in Figure 2, relying on an external model leads to three major issues. First, the automated metrics vary greatly depending on the type of the external model used for evaluation and they often have an arbitrary range, e.g., See Table 2,3 for the range of CLIP Radford et al. (2021) and MID Kim et al. (2022) scores. Second, many generative models rely on an external model such as CLIP's text encoding during training, and thus using the same CLIP model for automated evaluation biases the results. Finally, the external model itself can have several limitations (like poor performance on few image-text tasks for CLIP and hallucination for LLMs Xu et al. (2024)) making its scores unreliable.

SELFEVAL only uses the generative model and thus its scores directly reflect the strengths and weaknesses of the generative model. Note that most automated metrics operate on generated images whereas SELFEVAL uses real images from image-text recognition datasets.

## 2 Related Works

**Generative models**: Generative models learn to model the joint distribution, $p(X, Y)$ of data consisting of an observed variable $X$ and the target $Y$. The model can then be used to sample novel data. In this work, we are interested in image generation models.

Generative Adverserial Networks (GAN) Goodfellow et al. (2014); Radford et al. (2015), Variational AutoEncoders (VAE) Kingma & Welling (2014) and Denoising Diffusion Probabilistic models (DDPM) Ho et al. (2020) are some of the most popular image generation models in the literature. GANs belong to the category of generative models, where two distinct components, a generator and a discriminator, are pitted against each other within a zero-sum game framework. VAEs are a category of autoencoders

**Figure 2: Drawbacks of CLIP for generative model evaluation**. (Left) We compare the CLIP similarity scores of two latent diffusion models trained with CLIP ViT-L/14 (LDM-CLIP (ViT-L/14)) and OpenCLIP ViT-H/14 (LDM-CLIP (ViT-H/14)) text encoders. On the left, we compare the CLIP similarity scores, computed using CLIP ViT-L/14, on prompts from DrawBench, Winoground and, COCO datasets. The plot on the right compares the CLIP similarity scores computed using OpenCLIP ViT-H/14 model. The ranking changes depending on the model used. (Right) CLIP has poor performance in tasks of counting, spatial relationships, binding attributes and text corruption which constitute about 25% prompts in DrawBench. In each example, the correct caption is shown in green and CLIP picked the caption in bold. Using CLIP to evaluate text to image models on such prompts is not optimal.

that ensure "regularity" within the latent space by constraining their distribution to closely align with a well-behaved and typically standard normal distribution. In more recent times, DDPMs have exceeded the capabilities of all preceding state-of-the-art image generative models in terms of their generative prowess. Drawing inspiration from non-equilibrium statistical physics, Diffusion probabilistic models Sohl-Dickstein et al. (2015) employ a forward diffusion process to gradually destroy the structure in the unknown input distribution and transforming it into a well-behaved and tractable distribution. A reverse diffusion process is trained to learn to restore the structure, thereby learning the input distribution. An explicit connection between diffusion models and denoising score matching in Ho et al. (2020), enabled a simplified objective for training diffusion models. We utilize diffusion models in this study due to their outstanding image generation performance Dhariwal & Nichol (2021).

**Diffusion models**: In a relatively short time, diffusion models have surpassed GANs and VAEs as the defacto models for image generation due to their superior quality Dhariwal & Nichol (2021) and flexibility. Numerous studies have shown that diffusion models can be conditioned on a variety of modalities, including object classes Peebles & Xie (2023); Ho et al. (2020), natural language captions Saharia et al. (2022); Rombach et al. (2022); Nichol et al. (2022); Ramesh et al. (2022), camera pose Liu et al. (2023), images Brooks et al. (2023), bounding boxes Li et al. (2023b), segmentation, edge and depth maps Zhang & Agrawala (2023). Among these, text-conditioned diffusion models have attracted significant interest and popularity. Given paired image, caption data, these models are trained to fuse the caption features, extracted using a pre-trained text encoder, with the image features using cross attention. Text-to-Image (T2I) models demonstrate a remarkable comprehension of compositionality within text, often highlighted by their capacity to generate images based on counterfactual textual descriptions (like avacado shaped chair *etc.*). The most popular text encoders in use today are text encoders from the CLIP Radford et al. (2021) and the T5 Raffel et al. (2020) transformer. In this work, we analyze the text understanding of the diffusion models trained with different text encoders.

There exist two families of diffusion models in the literature, namely, pixel (PDM) Saharia et al. (2022); Ramesh et al. (2022) and latent diffusion (LDM) Rombach et al. (2022), differing primarily in the nature of input. The diffusion process in pixel diffusion is performed on pixels making these models computationally expensive. LDMs Rombach et al. (2022) operate on the autoencoder's latent space, balancing the computational constraints with the quality of PDMs. In this work, we analyze the text understanding of two state-of-the-art models with different text encoders from PDMs and LDMs.

**Classifiers with diffusion models**: Lately, there has been a increase in the usage of conditional diffusion models as classifiers, driven by their superior understanding of the conditioned modality. These models are surprisingly good at capturing intricate patterns and dependencies within the conditioning

input, making them strong discriminative models across a range of downstream tasks. Notable works include He et al. (2023), Mukhopadhyay et al. (2023) that either finetune a diffusion model, or use linear probing, for several classification and reasoning tasks. Unlike these methods, we do not train any models but instead convert the generative model into a discriminative one to understand its text understanding capabilities. Along similar lines to SELFEVAL, Clark & Jaini (2023); Li et al. (2023a) employ the ELBO loss as a proxy to estimate the likelihood scores (and subsequently the posterior using Bayes' rule) from diffusion models for several image understanding tasks. Li et al. (2023a)also report promising results on ITM tasks on the Winoground Thrush et al. (2022) dataset. Instead, we propose a systematic way, accounting for numerical stability, to estimate the likelihood of an image given the text from a conditioned diffusion model. To the best of our knowledge, SELFEVAL is the first method to show that the discriminative performance of generative models aligns closely with 'gold-standard' human evaluations in the assessment of generative models.

**Image-Text Matching for evaluating generative models**: CLIP R-precision Park et al. (2021) is a metric to evaluate the text understanding capabilities of generative models similar to SELFEVAL. While CLIP R-precision measures the text retrieval performance given a **generated** image, SELFEVAL uses the **ground truth** image-text pairs for evaluation. The generated images are often out-of-distribution to the external model, *i.e.* CLIP, making the score unreliable. SELFEVAL avoids this by using the generative model, instead of an external model, for evaluation.

## 3 Method: Converting generative models to discriminative models

Our method converts generative (diffusion) models into discriminative models by simply changing the inference, and does not require any retraining. This allows us to use the diffusion model itself on a variety of different image-text benchmarks and assess the diffusion model's image-text understanding capabilities. We briefly discuss an overview of diffusion models in Sec. 3.1 followed by our proposed method in Sec. 3.2

### 3.1 Preliminaries

Diffusion Probabilistic Models (DPM) Sohl-Dickstein et al. (2015) belong to a class of generative models trained to 'denoise' inputs created by a Markovian *forward* process. The forward process starts with a sample $\mathbf{x}_0$ and repeatedly adds Gaussian noise over $t$ timesteps to generate $\mathbf{x}_t$:

$$q(\mathbf{x}_t|\mathbf{x}_{t-1}) \sim \mathcal{N}(\mathbf{x}_t; \sqrt{1-\beta_t}\mathbf{x}_{t-1}, \beta_t \mathbf{I}). \tag{1}$$

Here $q(\mathbf{x}_0)$ is the data distribution. $\beta_t$ is the strength of the noise at timestep $t$ with $\beta_0 = 0, \beta_T = 1$. Note that $\mathbf{x}_t$ are the same size as the input. The joint distribution of the input along with the latents $q(\mathbf{x}_{0:T})$ is

$$q(\mathbf{x}_{0:T}) = q(\mathbf{x}_0) \prod_{t=1}^{T} q(\mathbf{x}_t|\mathbf{x}_{t-1}) \tag{2}$$

To sample images, one applies the *reverse* process, $p(\mathbf{x}_{t-1}|\mathbf{x}_t)$, starting with $\mathbf{x}_T$ sampled from the unit normal distribution, $\mathcal{N}(\mathbf{0}, \mathbb{I})$. So the joint distribution of the reverse process can be described as

$$p(\mathbf{x}_{0:T}) = p(\mathbf{x}_T) \prod_{t=1}^{T} p(\mathbf{x}_{t-1}|\mathbf{x}_t) \tag{3}$$

The reverse process $p(\mathbf{x}_{t-1}|\mathbf{x}_t)$ is not tractable and is often modeled using a neural network whose parameters are characterized by $\theta$, *i.e.* $p_\theta(\mathbf{x}_{t-1}|\mathbf{x}_t) \sim \mathcal{N}(\mathbf{x}_{t-1}; \boldsymbol{\mu}_\theta(\mathbf{x}_t, t), \boldsymbol{\Sigma}_\theta(\mathbf{x}_t, t))$.

## 3.2 Likelihood estimates from diffusion models

We specifically focus on text-to-image diffusion models, although our formulation extends to any conditional diffusion model. Text-to-image diffusion models are trained on a large datasets of image-text $(\mathbf{x}, \mathbf{c})$ pairs and model the reverse diffusion process $p(\mathbf{x}_{t-1}|\mathbf{x}_t, \mathbf{c})$. We 'invert' such a generative model and use it to estimate the likelihood of a real image $\mathbf{x}$ given a text caption $\mathbf{c}$, *i.e.*, $p(\mathbf{x}|\mathbf{c})$. We note that our method only changes the inference of a diffusion model and does not require any training. Assuming uniform prior on the classes, the likelihood $p(\mathbf{x}|\mathbf{c})$ can be converted into the posterior, $p(\mathbf{c}|\mathbf{x})$ using Bayes' rule, *i.e.* $p(\mathbf{c}|\mathbf{x}) = \frac{p(\mathbf{x}|\mathbf{c})}{|\mathcal{C}|}$, where $\mathcal{C}$ is the set of all classes.

Given the reverse process of a diffusion model parameterized by $\theta$, the likelihood for a datapoint $\mathbf{x}_0$ is

$$p_\theta(\mathbf{x}_0|\mathbf{c}) = \int p_\theta(\mathbf{x}_{0:T}|\mathbf{c})d\mathbf{x}_{1:T} \tag{4}$$

$$= \int p(\mathbf{x}_T)\prod_{t=1}^{T} p_\theta(\mathbf{x}_{t-1}|\mathbf{x}_t, \mathbf{c})d\mathbf{x}_{1:T} \tag{5}$$

Since the diffusion models reverse process $p_\theta(\cdot)$ is also a gaussian, we can further write this as

$$p(\mathbf{x}_0|\mathbf{c}) = \int p(\mathbf{x}_T)\prod_{t=1}^{T} \frac{1}{\sqrt{(2\pi)^D|\Sigma_\theta|}}$$
$$\times \exp\left(-\frac{1}{2}(\mathbf{x}_{t-1} - \boldsymbol{\mu}_\theta(\mathbf{x}_t, \mathbf{c}))^T \times \boldsymbol{\Sigma}_\theta^{-1}(\mathbf{x}_{t-1} - \boldsymbol{\mu}_\theta(\mathbf{x}_t, \mathbf{c}))\right)d\mathbf{x}_{1:T} \tag{6}$$

Here, $p(\mathbf{x}_T) \sim \mathcal{N}(0, \mathbb{I})$. For the sake of simplicity, we denote any realization of the random variable $\mathbf{x}_0$ as $\mathbf{x}_0$. Given a natural language caption $\mathbf{c}$, an image $\mathbf{x}_0$ and the noised latents $x_{1:T}$, the quantity inside the integral in Eq. 6 can be estimated numerically. We compute a Monte Carlo estimate of the integral by sampling $N$ noise terms ($\epsilon$) and computing $p(\mathbf{x}_0|\mathbf{c})$ as

$$p(\mathbf{x}_0|\mathbf{c}) = \sum_{n=1}^{N} p(\mathbf{x}_T^n)\prod_{t=1}^{T} p(\mathbf{x}_{t-1}^n|\mathbf{x}_t^n, \mathbf{c})$$
$$\text{where } \mathbf{x}_t^n = \sqrt{1 - \beta_t}\mathbf{x}_{t-1}^n + \sqrt{\beta_t}\epsilon^n \tag{7}$$

**Practical considerations:** The terms on the RHS of Eq. 7 are multivariate gaussians and analytically computing them involves exponentials which can be numerically unstable. This can be prevented by computing log probabilities instead. Taking log both sides of Eq. 7, we get

$$\log p(\mathbf{x}_0|\mathbf{c}) = \log\sum_{n=1}^{N} p(\mathbf{x}_T^n)\prod_{t=1}^{T} p(\mathbf{x}_{t-1}^n|\mathbf{x}_t^n, \mathbf{c}) \tag{8}$$

$$\geq \sum_{n=1}^{N}\left(\log p(\mathbf{x}_T^n) + \sum_{t=1}^{T}\log p(\mathbf{x}_{t-1}^n|\mathbf{x}_t^n, \mathbf{c})\right) \tag{9}$$

Where Eq. 9 follows from Jensen's inequality for concave functions, *i.e.* $\mathbb{E}(f(x)) \leq f(\mathbb{E}(x))$. All terms in Eq. 9 are analytically computable log probabilities of multivariate gaussians, offering numerical stability.

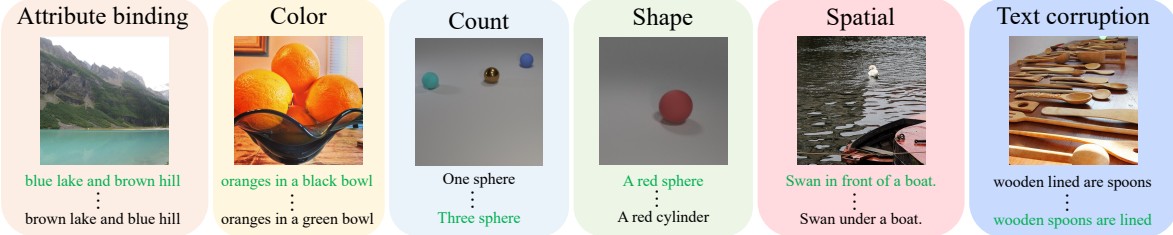

**Figure 3: Representative samples from the benchmark.** We divide the evaluation into six broad tasks, namely `Attribute binding`, `Color`, `Count`, `Shape`, `Spatial`, and `Text Corruption`. Each task is designed to evaluate a specific aspect of text faithfulness mimicking the categories in DrawBench. Each task is posed as an image-text matching problem, where given an image, the goal is to pick the right caption among distractors. The figure above shows examples from each task with the right caption highlighted in green.

**Implementation in practice:** Given an image $\mathbf{x}$ and a caption $\mathbf{c}$, we estimate $p(\mathbf{x}|\mathbf{c})$ as follows. As shown in Fig. 1, the noised latents $\mathbf{x}_{1:T}$ are computed using the forward process (Eq. 1). The final latent $\mathbf{x}_T$ is denoised for $T$ steps using the reverse process (the neural net) to obtain the denoised latents $\bar{\mathbf{x}}_{0:T-1}$. This process is repeated for $N$ independent noise vectors resulting in $\{\mathbf{x}_{1:T}^n\}_{n=1}^N, \{\bar{\mathbf{x}}_{0:T-1}^n\}_{n=1}^N$. Finally, the likelihood can be computed as $p(\mathbf{x}|\mathbf{c}) = \sum_{n=1}^N p(\mathbf{x}_T^n) \prod_{t=1}^T p(\mathbf{x}_{t-1}^n|\bar{\mathbf{x}}_t^n, \mathbf{c})$ (Eq. 9). We now show how estimating $p(\mathbf{x}_0|\mathbf{c})$ allows us to evaluate diffusion models

### 3.3 SelfEval to evaluate diffusion model's text faithfulness

The text faithfulness of a diffusion model measures its ability to understand the text prompt and ground it in the generated image output. The 'standard' way of evaluating text faithfulness uses a manually curated list of text prompts to generate images. The 'alignment' between the generated images and the text prompts can be measured using an external model or a human evaluator. The text faithfulness of a generative model inherently also measures its vision-language reasoning abilities. Thus, in SELFEVAL, we propose to directly measure the generative model's vision-language discriminative performance as a way to evaluate its text faithfulness.

We pose the SELFEVAL evaluation as an image-text matching problem and measure the generative model's recognition performance on standard discriminative image-text datasets. Thus, SELFEVAL does not rely on external models such as CLIP, does not need human evaluators, and does not need manual text prompt-set curation.

Image-text matching problems, such as image classification or retrieval, can be reformulated as picking the correct caption for a single image $\mathbf{x}$ from a set of captions $\{\mathbf{c}_i\}$. First, we can use a diffusion model to estimate $p(\mathbf{x}|\mathbf{c}_i)$ (described in Sec. 3.2) for each of the captions. The likelihood, $p(\mathbf{x}|\mathbf{c}_i)$, is then converted to the posterior, $p(\mathbf{c}_i|\mathbf{x})$, using Bayes' rule. Finally, the caption with the highest likelihood, *i.e.*, $\arg\max_{\mathbf{c}_i} p(\mathbf{c}_i|\mathbf{x})$, is chosen as the right one.

To compute the SELFEVAL score, we construct an evaluation set from a paired (image, caption) dataset. Each sample in this set consists of an image, its correct caption, and several distractor captions. We ensure that the distractor captions are sufficiently hard to increase task difficulty. For each image-caption set, we calculate the posterior probability using SELFEVAL and determine the model's accuracy, *i.e.* SELFEVAL score, over the entire set with the ground truth.

## 4 Experiments

We now use SELFEVAL to evaluate text-to-image diffusion models. In 4.1, we introduce our benchmark datasets and models, and present the SELFEVAL results in Section 4.2.

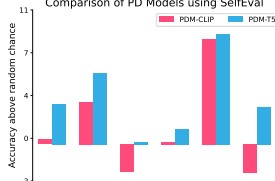 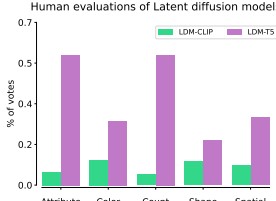 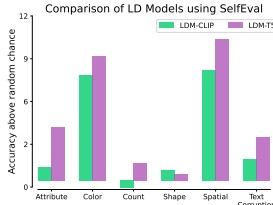

**Figure 4: Evaluating text-to-image models** using human evaluations and SELFEVAL. We evaluate different types of text-to-image models such as pixel diffusion (first two columns) and latent diffusion model (last two columns), and models that use different text encoders such as T5 XXL and CLIP. We observe that across all 4 diffusion models the relative ordering given by SELFEVAL's accuracy correlates with the pairwise human evaluation results. We also observe that latent diffusion models have a higher SELFEVAL accuracy than pixel diffusion models suggesting better text-faithfulness. Using the stronger T5 text encoder leads to better performance across human evaluations and SELFEVAL.

## 4.1 Benchmark and Evaluation

In SELFEVAL, we pose the text faithfulness evaluation as an image-text matching task, where the goal is to pick the right image caption pair among distractors.

**Tasks.** We identify six broad reasoning tasks for evaluation (Figure 3): 1) `Attribute binding`, 2) `Color`, 3) `Count`, 4) `Shape`, 5) `Spatial relationships`, and 6) `Text corruption`. Each of these tasks evaluate the model's understanding of a specific aspect of text faithfulness and is similar to the categories of prompts from DrawBench Saharia et al. (2022). The six tasks are constructed using data from TIFA Hu et al. (2023), CLEVR Johnson et al. (2016) and ARO Yuksekgonul et al. (2023).

**Datasets. TIFA** Hu et al. (2023) consists of 4000 text prompts, collected manually and from image captioning datasets, to evaluate the text faithfulness of generative models. In our evaluation, we use ∼2000 of these text-prompts that are constructed from the COCO Lin et al. (2014) dataset and convert the dataset from question-answering to an image-text matching format as detailed in the supplement. **Attribution, Relation and Order (ARO)** Yuksekgonul et al. (2023) is a benchmark that uses data from Visual Genome Krishna et al. (2017) for attribute and spatial relations, and COCO for ordering tasks. **CLEVR** Johnson et al. (2016) is a benchmark for compositional understanding and visual reasoning using synthetic images. We adopt the splits proposed by Lewis et al. (2022) for our case.

We divide the datasets among all the reasoning task as follows. For attribute binding, we combine samples from ARO (attribution) and CLEVR. For colors and counts, we use corresponding samples from TIFA and CLEVR. For shapes, we use samples from CLEVR. Data for spatial relationships is from TIFA, CLEVR and ARO (relations). The data for the text corruption task is from the ARO (order sensitivity) dataset. A sample of each task consists of an image and multiple text prompts and the performance on the task is the accuracy of pairing the image with the right caption.

We measure the performance of text-to-image generative models on the benchmark using the following evaluation methods.

**SelfEval (Ours)** is an automatic evaluation method and uses both the images and text from our benchmark introduced in Section 4.1. For each benchmark task, we randomly sample 1000 examples and evaluate the classification performance on them. We repeat this three times and the report the mean accuracy. We use 10 trials (*i.e.* $N = 10$) and perform diffusion for 100 steps (*i.e.* $T = 100$) for all the models. Refer to the supplement for ablation experiments on $N, T$.

**Human evaluations** are the gold standard for judging the performance of text-to-image models using pairwise comparsions. We present humans with generations from two models and ask them to vote for one of four choices: "both" the generations are faithful, "none" of them are faithful, or if only one of

the two images ("Image 1" or "Image 2") demonstrates fidelity to the given prompt. For simplicity, we only report votes where there is a clear preference for a model. We randomly pick 250 text prompts from each benchmark task as conditioning for human evaluation and the images are generated using DDIM Song et al. (2021) sampling, with 100 denoising steps. Note that unlike SELFEVAL, human evaluations do *not* use the real images from the benchmark tasks and the human evaluators only look at the generated images.

### 4.1.1 Models

We use models with different image representations: pixel diffusion models which directly use the pixel RGB values, and latent diffusion models where the image is projected into a latent space using an auto-encoder. We pick models trained with different text encoders within each class. This enables us to analyze the effect of text encoder on the final performance within each class.

**Diffusion models with CLIP text encoder.**

We employ a model trained with the OpenCLIP Ilharco et al. (2021) text encoder with a ViT-H/14 backbone for latent diffusion, accessed via an API containing open-sourced model weights. This model, trained on a public dataset of 5 billion images (excluding explicit material), outputs images of $512 \times 512$ resolution. For pixel diffusion, we use the architecture of DALL-E-2 Ramesh et al. (2022) in our experiments and train a model. This model uses a CLIP (ViT-L/14) text encoder, produces images of $64 \times 64$ resolution, and has a total of 4.2B parameters. It is trained for 2M steps on an internal image-text dataset (Internal-Dataset).

**Diffusion models with T5 text encoder.** We train a UNet model for latent diffusion, similar to Rombach et al. (2022), but with the CLIP text encoder replaced by a T5 XXL Raffel et al. (2020) text encoder. This model outputs images of $256 \times 256$ resolution. Trained on Internal-Dataset for 2M steps using a latent space with a $4\times$ downsampling factor, the model has a total of 5.8B parameters. We train a pixel diffusion model with 7.5B parameters, similar to Imagen Saharia et al. (2022), on $64 \times 64$ resolution inputs for 2M steps using the same data. Following this, we use a super-resolution model to upsample the output to $512 \times 512$. With the exception of the CLIP-based latent diffusion model Rombach et al. (2022), all the other models are trained for the same number of steps on the exact same data to ensure fair comparison.

## 4.2 Main results

We evaluate the four text-to-image models and report results in Figure 4. For SELFEVAL, we report the accuracy difference with the chance accuracy, since each of the tasks has a different degree of difficulty.

**Agreement between SelfEval and human evaluation.** In Figure 4, we evaluate four different diffusion models using both human evaluation and SELFEVAL. The human evaluation performance, measured via pairwise comparison, aligns with the ranking given by SELFEVAL for both pixel and latent diffusion models. To our knowledge, this is the first work to correlate the discriminative performance of generative models with human evaluation for text-to-image diffusion models across various models and tasks. The strong alignment between SELFEVAL and human raters suggests that SELFEVAL is a reliable and interpretable way for evaluating and comparing the text faithfulness of different diffusion models.

Next, we use SELFEVAL to further analyze the performance of diffusion models.

**Effect of the text encoder.** Comparing the different text-encoders used in Figure 4, we observe that diffusion models using the stronger T5 text encoder perform better on most tasks than the ones using the CLIP text encoder. The stronger performance of T5-based models holds for both human evaluations and SELFEVAL. The SELFEVAL results indicate that diffusion models using the CLIP-based encoders perform poorly on the `Count` task, even worse than random chance. For the `Text Corruption` task, which involves identifying a linguistically correct sentence among distractors with shuffled word order,

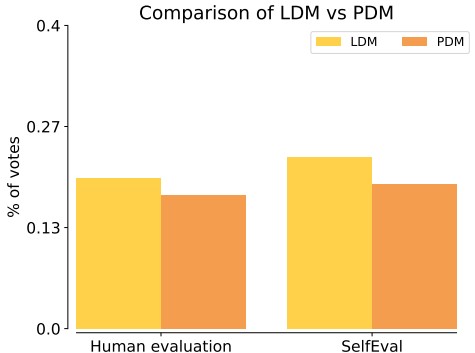

**Figure 5: Pixel vs Latent diffusion.** Human raters rank the generation of latent models higher than pixel models in text faithfulness. SELFEVAL exhibits a similar trend.

**Table 1: Diffusion models evaluated on the Winoground dataset**. We measure the image score (accuracy of picking the correct image given a text prompt) and text score (accuracy of picking the correct text given an image). Using SELFEVAL allows us to use diffusion models for both tasks unlike Li et al. (2023a) which leads to zero image score.

| Method | Model | Image Score | Text Score |
|---|---|---|---|
| CLIP (ViT-L/14) | − | 8.00 | 30.25 |
| OCLIP (ViT-H/14) | − | 12.75 | 30.75 |
| Li et al. (2023a) | LDM-CLIP | 0 | 34.00 |
| SELFEVAL | LDM-CLIP | 7.25 | 22.75 |
| SELFEVAL | PDM-CLIP | 14.00 | 17.00 |
| SELFEVAL | PDM-T5 | 12.00 | 28.25 |
| SELFEVAL | LDM-T5 | 13.50 | 29.00 |

the performance of CLIP-based models is lower. Thus, as suggested by prior work Yuksekgonul et al. (2023), CLIP models exhibit a bag-of-words understanding of the text.

**Pixel vs latent diffusion.** We compare the SELFEVAL performance of the pixel diffusion models to that of the latent diffusion models in Figure 5. Among models that use the same text encoder, *i.e.* PDM-T5 and LDM-T5, we observe that the latent diffusion models outperform the pixel diffusion ones in most cases, especially on the harder tasks of `Attribute Binding`, `Count`, `Spatial Relations` and `Text Corruption`. We hypothesize that this difference can be explained by the fact that the latent diffusion models operate on the compressed latent space and prioritize the text conditioning while 'offloading' the high-frequency image details to the autoencoder. We further investigate the performance of pixel and latent diffusion models by employing human raters to evaluate their text faithfulness in Figure 5. The data, for human evaluation, is constructed by randomly picking 500 examples from the all the tasks (100 examples from each task except text corruption), and choosing the right caption as the text prompt. We convert the accuracy of SELFEVAL, to votes, by counting the number of samples where only one model is right. From Figure 5, we observe that human raters prefer the generations of latent diffusion models over pixel diffusion models for text faithfulness. SELFEVAL also shows that latent diffusion models have a better text faithfulness and shows an alignment with human evaluations.

**Comparison with other metrics.** In Tables 2 and 3, we compare SELFEVAL with metrics that utilize external models on PDMs and LDMs, respectively. We compare SELFEVAL with the well-known CLIPScoreHessel et al. (2021) and recently proposed metrics such as MID Kim et al. (2022), LLMScore Lu et al. (2023), and VPEval Cho et al. (2023). In each table, we compute the scores for models using CLIP and T5 text encoders across five tasks (see Section 4.1) and compare them with human ratings. The row labeled "Human" in both tables indicates the winning votes obtained by each model. All rows use a green or red cell to denote whether the ranking among models with CLIP and T5 text encoders agrees or disagrees with human judgment, respectively. As mentioned in Section 1, we observe that the value ranges of MID vary wildly, making them uninterpretable and incomparable to the rest. CLIPScore measures the cosine similarity of the generated image with the input text prompt and has a range of $[-1, 1]$, while VPEval, LLMScore, and SELFEVAL compute accuracy. On the PDMs, we note that each metric disagreed with human judgment on at least one split. On the LDMs, we observe significant disagreement among different models. CLIPScore and SELFEVAL are the only two models with significantly less disagreement with human raters. In Fig. 6, we show the Spearman's rank correlation $\rho$ Spearman (1904) between each metric and the human ratings. On the x-axis, we plot $\rho$ for the PDM's results, and the LDM's results are plotted on the y-axis. The shaded region in the plot indicates the desired range for the correlations. Similar to Table 3, we observe that all metrics except SELFEVAL have a negative correlation with human ratings. SELFEVAL is the only

**Table 2: Comparison of evaluation metrics on pixel diffusion models.** We compare SELFEVAL with existing metrics on pixel diffusion models. "Humans" represents the winning votes obtained by the models in the human evaluation. Each row uses a green or red cell to denote whether the ranking agrees or disagrees with human judgment respectively. Note that the value ranges for CLIPScore and MID differ and are not directly comparable to those of the other metrics..

| Method | Attribute binding | | Color | | Count | | Shape | | Spatial | |
|---|---|---|---|---|---|---|---|---|---|---|
| | CLIP | T5 | CLIP | T5 | CLIP | T5 | CLIP | T5 | CLIP | T5 |
| Humans | 24 | 117 | 29 | 42 | 41 | 69 | 30 | 7 | 19 | 72 |
| CLIPScore (↑) | 0.90 | 0.98 | 0.89 | 0.91 | 0.81 | 0.76 | 0.86 | 0.82 | 0.76 | 0.78 |
| MID (↓) | -8.6E14 | -2.1E14 | -2.7E5 | -1.8E5 | -8.6E3 | -1.1E4 | -4.6E3 | -2.6E3 | -1.0E15 | -5.6E15 |
| VPEval (↑) | 61.6 | 63.1 | 86.9 | 86.0 | 16.7 | 31.9 | 94.6 | 91.9 | 18.1 | 25.3 |
| LLMScore (↑) | 6.6 | 8.2 | 15.5 | 15.8 | 7.3 | 9.2 | 10.82 | 9.9 | 23.5 | 25.6 |
| SELFEVAL (↑) | 50.4 | 53.3 | 28.5 | 30.8 | 22.8 | 25.2 | 33.2 | 34.3 | 33.6 | 34.0 |

**Table 3: Comparison of evaluation metrics on latent diffusion models.** We compare SELFEVAL with existing metrics on latent diffusion models. "Humans" represents the winning votes obtained by the models in the human evaluation. Each row uses a green or red cell to denote whether the ranking agrees or disagrees with human judgment respectively. We observe significant disagreement with human ratings for metrics, except SELFEVAL and CLIPScore, compared to results on pixel diffusion models.

| Method | Attribute binding | | Color | | Count | | Shape | | Spatial | |
|---|---|---|---|---|---|---|---|---|---|---|
| | CLIP | T5 | CLIP | T5 | CLIP | T5 | CLIP | T5 | CLIP | T5 |
| Humans | 14 | 140 | 27 | 69 | 11 | 140 | 25 | 48 | 21 | 73 |
| CLIPScore (↑) | 0.89 | 0.99 | 0.85 | 0.80 | 0.76 | 0.80 | 0.76 | 0.86 | 0.75 | 0.83 |
| MID (↓) | -8.1E14 | -1.3E14 | -1.1E5 | -1.1E5 | -2.1E4 | -7.6E3 | -1.1E3 | -2.1E3 | 4.33E14 | -8.0E15 |
| VPEval (↑) | 60.4 | 66.0 | 87.5 | 85.7 | 64.7 | 18.9 | 92.9 | 92.2 | 34.1 | 23.4 |
| LLMScore (↑) | 9.2 | 9.1 | 15.4 | 17.3 | 9.6 | 8.4 | 11.6 | 10.6 | 21.4 | 23.1 |
| SELFEVAL (↑) | 51.0 | 54.1 | 33.0 | 34.4 | 24.4 | 26.3 | 33.8 | 33.5 | 33.4 | 35.8 |

metric that correlates well with both PDMs and LDMs. Note that while all existing metrics compute the score on generated images, our proposed metric, SELFEVAL, uses the ground truth image-text pair to compute text faithfulness. These findings underscore the robustness of SELFEVAL in aligning with human judgment across different models and tasks, highlighting its potential as a reliable metric.

**Qualitative results.** Figure 7 (Top) compares the generations of pixel diffusion models that use T5 and CLIP text encoders. In each example, the image on the left and right are generated using CLIP and T5 text encoder respectively. We notice that as the difficulty of prompts increases, models with a stronger text encoder performs better. Both the models fail on the much harder task of counting instances and spatial relationships. In Figure 7 (Bottom), each example consists two images generated using a pixel diffusion model (left) and a latent diffusion model (right) with a T5 text encoder. We observe that unlike the pixel diffusion model, the latent diffusion model can get small yet important details right ("gray table cloth" and "white handles" in the second and sixth example respectively). We believe that the latent diffusion model can offload the high frequency appearance details to the autoencoder, allowing it to pay more attention to the conditioning variable.

### 4.3 Generative models applied to other reasoning tasks

We now use the challenging Winoground Thrush et al. (2022) benchmark to evaluate the vision-language reasoning abilities of diffusion models. Winoground defines two tasks - (1) 'text score' that evaluates text retrieval given an image; and (2) 'image score' that evaluates image retrieval given a text prompt.

**SelfEval vs concurrent work** Concurrent work from Li et al. (2023a) demonstrates that diffusion models perform well on the Winoground text score task and achieve competitive performance with

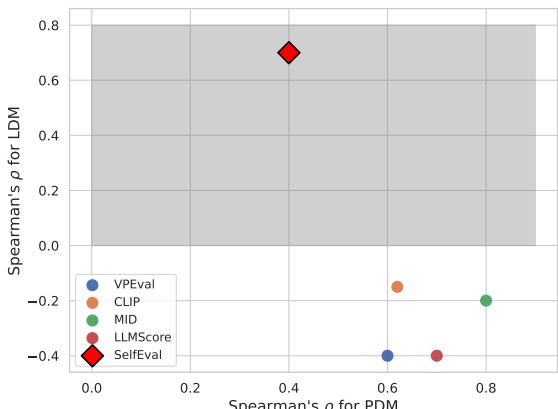

**Figure 6: Spearman's correlation with ground truth across evaluation metrics.** We compute the Spearman rank correlation between human ratings and metrics in Tables 2, 3. Existing metrics show good correlation with pixel diffusion models but a negative correlation with latent diffusion models. In contrast, SELFEVAL positively correlates with both types of models.

**Table 4: Drawback of MID Kim et al. (2022).** We compute MID using two CLIP models of two LDM-CLIP models with different backbones. Like CLIP Score, MID is sensitive to the CLIP model used for evaluation.

| Model | MID↓ | |
|---|---|---|
| | ViT-B/32 | ViT-L/14 |
| LDM-CLIP (ViT-L/14) | **27.77** | 25.70 |
| LDM-CLIP (ViT-H/14) | 29.25 | **23.53** |

**Table 5: Performance of CLIP on the benchmark.** We evaluate the zero-shot performance of CLIP (ViT-L/14) on the six tasks. "Random" is the chance accuracy. CLIP achieves impressive performance on the tasks of `Color` and `Shape`. The performance of CLIP is close to random on the remaining tasks making it unsuitable for evaluating generative models on such prompts.

| Model | Attribute binding | Color | Count | Shape | Spatial | Text corruption |
|---|---|---|---|---|---|---|
| Random | 50 | 25 | 25 | 33 | 25 | 20 |
| CLIP | 55.40 | 85.20 | 67.80 | 91.10 | 40.50 | 51.00 |

discriminative models. Using their formulation yields poor results (zero accuracy) on the image score task as shown in Table 1. Li et al. (2023a) use the ELBO loss as a proxy for the likelihood $p(\mathbf{x}|\mathbf{c})$ which works well for comparing different text prompts and thus leads to good text score performance. However, our analysis revealed that the ELBO loss computed for the predictions from two different images are not comparable, and thus leads to zero image score. SELFEVAL on the other hand, doesn't approximate the likelihood but instead estimates it as described in Sec 3. Using SELFEVAL leads to a non-zero image-score for the same generative model used by Li et al. (2023a), and yields performance close to that of the discriminative CLIP ViT-L model.

**SelfEval applied to other diffusion models.** Using SELFEVAL reveals that all the diffusion models introduced in Section 4.1.1 achieve competitive performance on both the image score and text score tasks. Compared to all the discriminative CLIP models, generative models achieve strong results in both image and text scores using SELFEVAL. This result reinforces the notion that optimizing the generative objective can provide non-trivial and complementary improvements for several visuo-linguistic reasoning tasks. For additional analysis on various hyperparameters on the performance on Winoground, refer to the supplement.

## 4.4 Drawbacks of existing metrics

In this section, we discuss limitations of metrics computed using external models like CLIP and LLMs that SELFEVAL can effectively address. CLIP score, the most common metric for evaluating text faithfulness of generative models, measures the cosine similarity between the features of the generated image and the conditioned text caption. Recently, Mutual Information Divergence (MID) Kim et al. (2022) used CLIP features to compute the negative Gaussian cross-mutual information between the image and text prompt. LLMScore Lu et al. (2023) evaluates the text faithfulness of text-to-image models by prompting an LLM to generate a score and rationale given a generated image and a text prompt. VPEval Cho et al. (2023) is an interpretable and explainable metric for text-to-image generation based on visual programming Gupta & Kembhavi (2022)Surís et al. (2023).

**Sensitivity to the exact CLIP model.** We report the CLIP similarity scores of the generations from two versions of the Latent Diffusion Models Rombach et al. (2022) on prompts from DrawBench Saharia

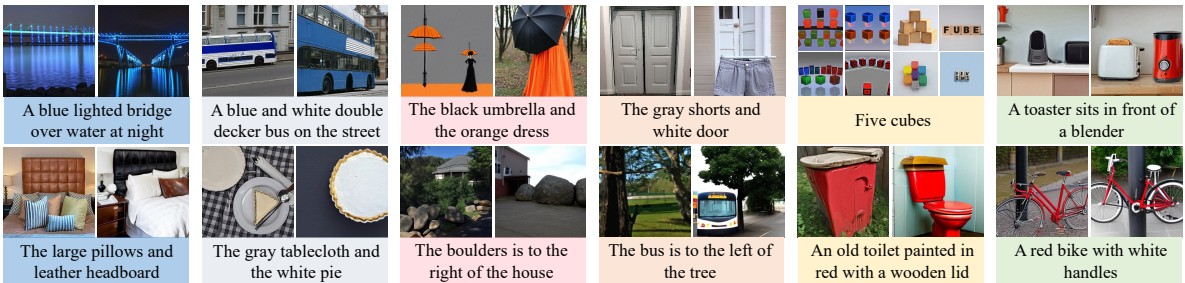

**Figure 7: Qualitative Results.** (Top): Each example compares the generations of pixel diffusion models with CLIP (left) and T5 (right) text encoders. As the difficulty of the prompt increases, models with stronger text encoders maintain higher text fidelity. Both the models fail on simple prompts from `Count` and `Spatial relationships`. (Bottom): Comparison between generations of Pixel (left) and Latent (right) diffusion models with a T5 text encoder. Latent diffusion models can get smaller details like "gray cloth" and "white handles" (second and last example respectively) correctly.

et al. (2022), Winoground Thrush et al. (2022) and COCO-minival Lin et al. (2014) datasets in Figure 2. The first model (LDM-CLIP (ViT-L/14)) uses the text encoder of CLIP with ViT-L/14 backbone and the second model (LDM-CLIP (ViT-H/14)) uses the text encoder with OpenCLIP Ilharco et al. (2021) ViT-H/14 visual backbone. Across all the three datasets, we observe that LDM-CLIP (ViT-L/14) ranks higher than LDM-CLIP (ViT-H/14) if a CLIP (ViT-L/14 visual backbone) model is used, but ranks lower with an OpenCLIP (ViT-H/14 visual backbone). Our hypothesis is that images generated by a model using a particular CLIP text encoder may still contain some residual information, which could cause them to receive higher scores when assessed using the same CLIP model. This type of bias was identified by Park et al. (2021) in the context of evaluation of text-to-image models, though not in relation to the CLIP score. We observe that this bias is not limited to the CLIP score but pertains to any metric that uses CLIP features.

Table 4 shows the MID Kim et al. (2022) computed using CLIP ViT-B/32 and ViT-L/14 backbones of two models, LDM-CLIP (ViT-L/14) and (ViT-H/14) on prompts from the Winoground Thrush et al. (2022) dataset. Our observations show that when MID is computed using ViT-B/32, LDM-CLIP (ViT-H/14) ranks higher than LDM-CLIP (ViT-L/14). Conversely, when MID is computed using the ViT-L/14 backbone, LDM-CLIP (ViT-L/14) ranks higher than LDM-CLIP (ViT-H/14). We emphasize the need for caution among researchers who employ this metric, particularly concerning this bias. SELFEVAL avoids this problem as we do not employ an external model for evaluation.

**CLIP score is limited by CLIP's performance** and thus using it as a proxy on tasks where CLIP itself has poor performance does not yield meaningful comparsions. While the CLIP model has demonstrated impressive zero-shot performance on several image-text tasks, it has severe limitations on many complex reasoning tasks. We compute the performance of CLIP ViT-L/14 model on the six tasks introduced in Section 4.1 and report the results in Table 5. CLIP performs well on `Color` and `Shape` but its performance on all the other tasks is poor. On the widely used DrawBench prompts, 25% of the captions evaluate the generations for attribute binding, counting, spatial relationships and text corruption. Thus, using CLIP to evaluate generations on such prompts in DrawBench is not ideal. SELFEVAL avoids this problem by directly leveraging the diffusion model itself.

**Hallucination in LLMs affects evaluation**. VPEval Cho et al. (2023) and LLMScore Lu et al. (2023) employ vision foundation models to ground the concepts, mentioned in the text prompt, in the generated image and leverage a LLM to verify the prompt adherence. This would work if the LLMs are robust, but as shown in Fig. 8, LLMs hallucinate irrelevant information making the evaluation unreliable. This is reflected in the worse than chance performance of LLMScore Lu et al. (2023) in Tables 2, 3 across all the metrics making them unsuitable for such evaluation tasks. We reiterate

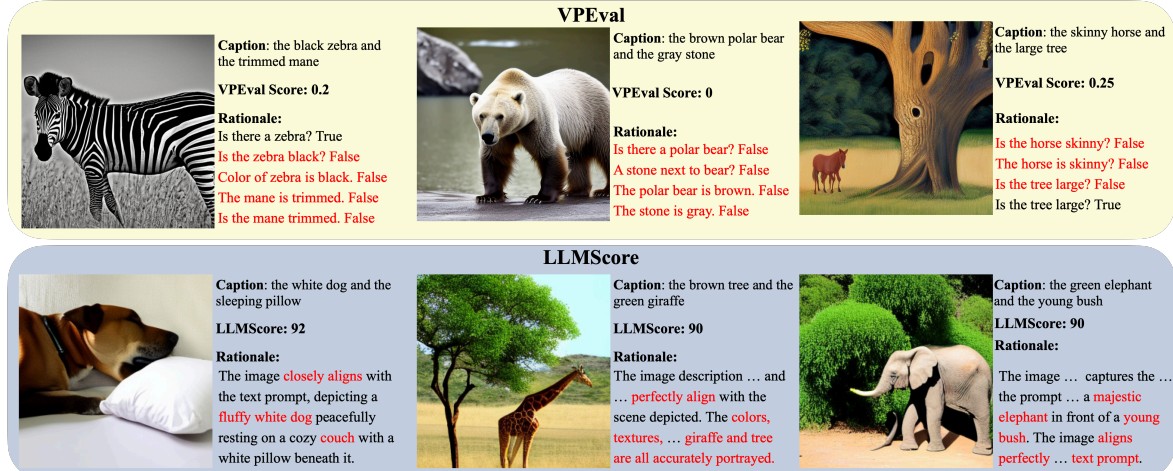

**Figure 8: Hallucination** in large language and vision models affects evaluation of generative models. (Top): Three examples showing the caption, scores and corresponding rationale from VPEval Cho et al. (2023). Wrong entries are highlighted in red. We observe that VPEval penalizes the generation despite its faithfulness to the text, due to hallucinations of the vision and language model. (Bottom): Three examples showing the generated image, a wrong caption and the rationale generated by LLMScore Lu et al. (2023). LLMScore assigns a high score despite the incorrect caption, due to the hallucinations of the LLM, as highlighted in red in the rationale.

that the performance of the external model on the evaluation set has a large impact on its evaluation capabilities and SELFEVAL is one such way to eliminate the reliance on external models for evaluation.

## 5 Conclusion and Limitations

This paper introduced SELFEVAL, an automated method for evaluating the text-understanding capabilities of diffusion models. SELFEVAL estimates the likelihood of real images given text prompts using the diffusion model itself, eliminating the need for external discriminative models. Our experiments demonstrated that SELFEVAL aligns with human evaluations across various models and tasks, proving its reliability as an automated metric for text-conditioned image generation. We anticipate that such metrics will expedite diffusion model research and encourage further improvements. SELFEVAL's applicability extends beyond text-to-image diffusion models, potentially serving in the evaluation of other conditioned diffusion models like text-to-audio and text-to-video. However, in its current form, SELFEVAL is designed to work with generative modeling methods that estimate likelihood, and while there are workarounds, they are not trivial. Additionally, we assume that the generative model has been trained sufficiently, as the overall SELFEVAL score inherently depends on the training quality of the generative model itself. Future work aims to generalize SELFEVAL for use with non-diffusion-based generative models.

## 6 Broader Impact

This research introduces SELFEVAL, an automatic evaluation method, eliminates the use of an external model for evaluation while demonstrating strong agreement with human annotators. This automated evaluation could significantly reduce the monetary and time costs associated with hiring human evaluators, making the process of developing and refining generative models more efficient. Progress in research areas related to SELFEVAL could potentially improve the ability of generative models to adhere to text. However, such models pose a serious concern as they could lead to the creation of deceptive images if placed in the wrong hands.

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
