# OpenReview forum: "SelfEval: Leveraging discriminative nature of generative models for evaluation"
_TMLR — Accepted by TMLR_

### Review · Reviewer_H2un · 2024-09-02

**Summary Of Contributions:**

The authors present an automated approach for evaluating the text alignment of text-to-image generative diffusion models using standard image-text recognition datasets. Their method, referred to as SelfEval, utilizes the generative model to compute the likelihood of real images given text prompts, which can then be employed to perform recognition tasks with the generative model.

The authors evaluate the generative models on standard datasets designed for multimodal text-image discriminative learning, assessing fine-grained aspects of their performance, including attribute binding, color recognition, counting, shape recognition, and spatial understanding.

Unlike existing automated metrics that rely on an external pretrained models which are sensitive to the specific pretrained model and its limitations, SelfEval circumvents these issues. If the authors' claims are true, this is the first automated metric that demonstrates a high degree of agreement with gold-standard human evaluations for measuring text-faithfulness across multiple generative models, benchmarks, and evaluation metrics.

Additionally, SelfEval reveals that generative models exhibit competitive recognition performance on challenging tasks such as Winoground image-score when compared to discriminative models.

**Audience:**

No

**Broader Impact Concerns:**

No concerns yet.

**Claims And Evidence:**

No

**Requested Changes:**

1. The authors should explain with more details about how SelfEval works. Is it only works for the fixed classes (captions)? What is the prior distribution over the classes? If so, what is the prior distribution over the captions? Uniform? If their result is not limited to the classification over fixed captions, how one could define the prior distribution? How did you generate and select distractors in Figure 3? I definitely feel that they should explain their general framework, much more than the things in Section 3.3.

2. Also, the authors should also explain how to measure the closeness of the captions generally, before the start of the experiment section. For example, suppose that in Figure 3, Red sphere is quite close to Red cylinder, but not close to blue tiger (for example). Do you measure the closeness between two captions? Or regarding it as 0-1 (1 if correct, 0 otherwise)? If the details change over tasks, then the author should clarify what is 'common structure' of SelfEval before the experiment, and what can be fine-tuned.

**Strengths And Weaknesses:**

Clarity: The "Preliminary" and "Related works" sections are written in sufficient detail that even non-experts can understand them. I believe this provides a good explanation for beginners in the field. However, the explanation of how "SelfEval" actually works is severely lacking. If I understand correctly, the only structural explanation of SelfEval is in Section 3.3 on page 6, and even this does not make it clear how SelfEval intends to perform evaluations.

Novelty and Soundness: As I understand it, this paper ultimately proposes using the reverse process of a generative model to estimate
$p(x∣c)$, and then using the highest likelihood to perform classification. Then it measures the 'matching performance' of each generative model. While the abstract is written in a grandiose manner, the summary in Section 3.3 suggests that this SelfEval works only with finite number of captions. As a reviewer who is not highly experienced in experiments in this field, I would appreciate a clear summary before the experiment explanation, particularly regarding what happens if the image is not part of the set of captions, or whether the classification is only for images within the set of captions.

---

> ### Author Response · Authors · 2024-11-22
> **Response to Reviewer H2un's comments**
>
> We thank the reviewer for their valuable comments on the merits of SelfEval, on the paper writing quality of a specific sections and experimental results of our work. We address your comments below.
>
> **Clarity**: We thank the reviewer for pointing this out. We will explain the working of SelfEval here and will update the manuscript to reflect this. We aim to evaluate the text faithfulness of generative models by formulating this task as image-text matching. The objective is to identify the correct caption from a set of distractors for a given ground truth image. This involves computing the accuracy of the generative model on ground truth image-text pairs. To achieve this, we utilize standard image-text datasets and calculate accuracies based on them. Specifically, given a ground truth image and N captions, we determine the accuracy of the generative model in selecting the correct caption.
> As described in Section 3.3, we use the likelihood score calculated using Equation 9 to compute the posterior probability via Bayes' rule. The posterior $p(c|x)$ represents the image classification score. We select the caption with the highest score, $\text{argmax}_{c_i}p(c_i|x)$
> as the correct one and then compute the accuracy. We hope this explanation helps to clarify how SelfEval works for the reviewer.
>
> **Novelty and Soundness**: SelfEval is not restricted to working with a finite number of captions. Similar to any classification model, we evaluate the accuracy of a generative model in an N-way classification task. In this work, we limit the value of N $\in \{ 4, 5 \}$. However, SelfEval is capable of functioning for any N greater than 5. We ensure that we select the top $N-1$ hard negatives as distractors, making the task as challenging as possible. We direct the reviewer to Table 5, where we demonstrate the performance of a powerful CLIP model on this dataset. Notably, CLIP achieves performance close to random in 3 out of 6 cases.
> We assess the classification performance of generative models using SelfEval. It is important to note that we **do not** intend to evaluate its out-of-domain detection (OOD) performance. Therefore, we assume that one of the N captions is correct i.e. the classification is only for images within the set of captions. We will include this clarification in the manuscript.
>
> **Requested changes**
> 1. We assume a uniform prior over the captions. Since we used hard negatives as distractors, we believe a uniform prior is the appropriate choice. Some hard negatives were provided by the dataset creators, while others were selected based on CLIP scores. We will elaborate on this before the experiments section, as suggested.
> 2. Thank you for your insightful feedback. We treat this as a 0-1 evaluation task, where we select hard negatives as distractors and compute accuracy without considering the closeness of the correct caption to the distractors. We believe this approach provides the most challenging yet fair evaluation of classification models. This setting is consistently applied across all different splits. We will incorporate this clarification into the manuscript.
>
> We will add this discussion to the main paper and we hope it answers all the reviewers' concerns.

---

### Review · Reviewer_WmNq · 2024-09-30

**Summary Of Contributions:**

The paper proposed SeflEval, an approach to evaluate the text-to-image alignment of generative models. The key idea is to use the generative model itself to perform this evaluation. It does this by computing the likelihood of the text prompt on real images rather than generated images. Extensive experiments were conducted. Results showed that SelfEval shows more text faithfulness than using CLIP scores and also more aligned with human evaluation.

**Audience:**

Yes

**Broader Impact Concerns:**

No concerns

**Claims And Evidence:**

Yes

**Requested Changes:**

Can you comment on how well the likelihood scores are calibrated across different generative models? This is critical for comparing different generative models (e.g., pixel vs. latent models, as shown in the paper) with each other. Is there a way to verify if the score is well-calibrated or not?

**Strengths And Weaknesses:**

**Strengths**
- Not having to rely on external models for evaluation is nice. Also automated evaluation as opposed to requiring human intervention. This gives us a lot of flexibility.
- Experiments show the benefits of SelfEval, including attribute binding, color and shape recognition, spatial relations, and text corruption.
- Results show good consistency between SelfEval and human evaluation.

**Weaknesses**

The paper is solid and does a thorough job of evaluating SelfEval.  I did not find any significant weaknesses.

---

> ### Author Response · Authors · 2024-11-21
> **Response to Reviewer WmNq's comments**
>
> We thank the reviewer for appreciating the motivation of the paper, the merits and results of SelfEval.
>
> **Requested Changes**
> Yes! That is an excellent point. We observed that the likelihood scores of pixel diffusion models differ from those of latent diffusion models, and these scores need to be properly calibrated to ensure a fair comparison. However, we do not directly compare the scores. Instead, we compute classification accuracy (ranging from 0-100%), which is well-calibrated, to compare different models. Given an image and N captions, we compute N likelihood scores (which are comparable since they are output from the same model) and use these to determine the model's accuracy in predicting the correct image-caption pair. Thus, the comparison occurs after computing the accuracy. This approach ensures that the SelfEval scores are well-calibrated and that the comparisons are fair.
> We will add this to the manuscript.

---

### Review · Reviewer_fHRR · 2024-11-01

**Summary Of Contributions:**

This paper introduces SelfEval, a method to evaluate text-to-image generative models by converting them into discriminative models. SelfEval assesses a model's text-faithfulness and visual understanding without relying on external models like CLIP, which the authors argue introduces biases and limitations. The method computes the likelihood of real images given text prompts and evaluates the model’s performance across tasks like color recognition, counting, and spatial understanding. SelfEval demonstrates a high correlation with human evaluations, potentially offering a scalable and objective evaluation tool for generative models.

**Audience:**

Yes

**Broader Impact Concerns:**

The broader impact has been discussed in the paper.

**Claims And Evidence:**

Yes

**Requested Changes:**

1. How well does SelfEval handle more complex spatial relationships or compositions? It would be helpful to see some analysis on this front to understand how it performs in scenarios where there’s high visual-text alignment complexity.

2. Could SelfEval be adapted to assess image quality in addition to text-faithfulness? Right now, the method focuses only on text-faithfulness, so it would be interesting to know if it could also be modified to evaluate overall image quality.

3. Since SelfEval’s approach works best with models that compute likelihoods directly (like diffusion models), does this limit its applicability to other generative models, such as GANs, that don’t have a straightforward likelihood-based mechanism?

**Strengths And Weaknesses:**

### Strengths
1. A major plus is that SelfEval doesn’t rely on any external models, so it avoids the dependencies and potential biases that external tools might introduce.
2. The authors have run extensive experiments, and the results show that SelfEval’s evaluations are closely aligned with human judgments across various tasks. This really boosts its reliability as a tool for model assessment.
3. SelfEval digs deep, evaluating specific tasks like color recognition, spatial relationships, and counting. This approach provides detailed insights into how well a model understands the visual-text alignment, which is super valuable.
4. Another strong point is that SelfEval is flexible, it’s built to work with any generative model that can calculate likelihoods, so it’s broadly applicable across different text-to-image models.

### Cons
1. One potential limitation here is that if the generative model itself isn’t well-trained or is being applied to unfamiliar tasks, its accuracy and reliability in SelfEval could suffer. This means the scores might not truly reflect the model’s performance on such tasks, which could impact the reliability of the evaluation.

---

> ### Author Response · Authors · 2024-11-21
> **Response to reviewer fHRR's comments**
>
> We appreciate the reviewer for acknowledging the motivation and various merits of SelfEval. We also thank them for agreeing that our claims are well-supported and that there is an audience for our work. We address your concerns below.
>
> **Cons**: We would like to highlight that SelfEval is a principled method for transforming a generative model into a discriminative one, or classifier, to assess its text faithfulness. Similar to image classifiers, the reliability of a generative model—when converted into a classifier using SelfEval—on out-of-domain tasks is not reliable. Typically, classifiers experience a drop in performance on unfamiliar tasks, and this is also true for a generative model converted into a classifier using SelfEval. Therefore, on unfamiliar tasks, we expect the generative model to achieve low accuracy, correctly indicating that the model is not well-suited for the unknown task.
>
> **Requested Changes**
> 1. For evaluations, we pick the hard negative captions as distractors for evaluation. We request the reviewer to provide more details regarding this analysis.
> 2. That is infact very interesting! However, designing automatic metrics to measure image quality is challenging. Existing metrics, such as FID, often do not align well with human judgment [1]. We believe that visual quality is highly subjective, and obtaining ground truth annotations for aspects like pixel sharpness and aesthetics is difficult due to ambiguity. What one person finds aesthetically pleasing, another may not. Additionally, visual quality heavily depends on the quality of the training data.
> With noisy ground truth data, the scores obtained by SelfEval are not reliable. However, SelfEval can be used to measure more objective qualities, such as style (e.g., anime, impressionist, cartoon), without any modifications.
> To address the reviewers' question, SelfEval can be used to evaluate any objective aspects of visual quality without any modifications.
> 3. Yes! the reviewer is correct that SelfEval cannot be directly applied to cases like GANs, but there are workarounds. There is extensive literature on GAN inversion[2], which can be utilized to find latent vectors for different captions given a ground truth image. These latent vectors can then be passed through the generator to produce different versions, and the discriminator can be used to estimate the likelihood of the generated image given a caption. While this approach is not theoretically backed, we believe such a simple workaround can be employed for methods that do not directly estimate the likelihood score.
> We would like to emphasize that the specific method for generating classification scores may vary, but the core idea of SelfEval—that a generator can be used to evaluate itself—is very valuable and has several merits, as highlighted in this work
>
>
> [1] Jayasumana, Sadeep et al. “Rethinking FID: Towards a Better Evaluation Metric for Image Generation.” 2024 IEEE/CVF Conference on Computer Vision and Pattern Recognition (CVPR) (2023): 9307-9315.
> [2] W. Xia, et al., "GAN Inversion: A Survey," in IEEE Transactions on Pattern Analysis and Machine Intelligence, vol. 45, no. 3, pp. 3121-3138, 1 March 2023, doi: 10.1109/TPAMI.2022.3181070.

---

### Decision · Action_Editor_KWrQ · 2024-12-18

**Recommendation:** Accept as is

**Comment:**

The idea of the paper is simple, yet practical. This idea could be potentially applied to other alignment problems besides text-to-image. It allows evaluating text-to-image models without relying on external pre-trained models, whose biases and limitations could affect the accuracy of the evaluation. The authors support their proposed method, SelfEval, using a large set of experiments.

There are a few issues raised by the reviewers that are important to be addressed. I do no think it is needed to accept the paper with minor revision, but I strongly encourage the authors to try their best to address them.

1) Dedicate a paragraph or a section to list/highlight the limitations of SelfEval, including (i) it works with models that directly compute likelihood, e.g., diffusion models; (ii) if the generative model is not well-trained, its evaluation could be questionable; (iii) its success depends on how well the discriminative task is designed/selected (e.g., making sure to have hard negatives ...).

2) Better and more clearly explain how SelfEval works. This is the main contribution of the paper and the section dedicated to it (Section 3.3) is not well-written/self-contained. It is important to explain how the likelihood is computed. You briefly explain it in both Sections 3.2 and 3.3, why? Why not just explain it in one place and clearly mention how the likelihood is computed. It is also important to clarify how the discriminative task is selected and how the evaluation metric is computed. Perhaps a simple example of a discriminative task can help.

**Audience:**

This paper could be of interest to those who are working on text-to-image alignment (and possibly similar alignment problems).

**Claims And Evidence:**

In this paper, the authors propose a method, called SelfEval, to evaluate text-to-image alignment in generative diffusion models. Given a text-to-image discriminative learning problem, SelfEval evaluates the alignment of a text-to-image generative diffusion model by using it to compute the likelihood of real images given text prompts. This way SelfEval does not rely on external pre-trained models, such as CLIP, for its evaluation, and thus, is not sensitive to the biases and limitations of these models. The authors show experimentally that SelfEval has a high degree of agreement with human in measuring text-faithfulness.